# Can CRP Levels Predict Infection in Presumptive Aseptic Long Bone Non-Unions? A Prospective Cohort Study

**DOI:** 10.3390/jcm10030425

**Published:** 2021-01-22

**Authors:** Theodoros H. Tosounidis, Colin Holton, Vasileios P. Giannoudis, Nikolaos K. Kanakaris, Robert M. West, Peter V. Giannoudis

**Affiliations:** 1Department of Orthopaedic Surgery, University Hospital of Heraklion, Crete PC 71110 Heraklion, Greece; ttosounidis@yahoo.com; 2Academic Department of Trauma & Orthopaedics, School of Medicine, University of Leeds, Leeds LS2 9JT, UK; colin.holton@nhs.net (C.H.); vgiannoudis@aol.com (V.P.G.); n.kanakaris@nhs.net (N.K.K.); 3Leeds Institute of Health Sciences, University of Leeds, 101 Clarendon Road, Leeds LS2 9JT, UK; R.M.West@leeds.ac.uk; 4NIHR Leeds Biomedical Research Center, Chapel Allerton Hospital, Leeds LS7 4SA, UK

**Keywords:** long bone, nonunion, infection, low-grade, CRP, WBC

## Abstract

Nonunion remains a major complication of the management of long bone fractures. The primary aim of the present study was to investigate whether raised levels of C-reactive protein (CRP) and white blood cell count (WBC), in the absence of clinical signs, are correlated with positive intraoperative tissue cultures in presumptive aseptic long-bone nonunions. Infection was classified as positive if any significant growth of microorganisms was observed from bone/tissue samples sent from the theater at the time of revision surgery. Preoperatively all patients were investigated with full blood count, white blood count differential as well as C-reactive protein (CRP). A total of 105 consecutive patients (59 males) were included in the study, with an average age of 46.76 years (range 16–92 years) at the time of nonunion diagnosis. The vast majority were femoral (56) and tibial (37) nonunions. The median time from the index surgical procedure to the time of nonunion diagnosis was 10 months (range 9 months to 10 years). Positive cultures revealed a mixed growth of microorganisms, with coagulase-negative *Staphylococcus* (56.4%) being the most prevalent microorganism, followed by *Staphylococcus*
*aureus* (20.5%). *Pseudomonas*, Methicillin-Resistant Staphylococcus aureus (MRSA), coliforms and micrococcus were present in the remainder of the cases (23.1%). Overall, the risk of infection with normal CRP levels (<10 mg/L) was 21/80 = 0.26. Elevated CRP levels (≥10 mg/L) increased the risk of infection to 0.72. The relative risk given a positive CRP test was RR = 0.72/0.26 = 2.74. Overall, the WBC count was found to be an unreliable marker to predict infection. Solid union was achieved in all cases after an average of 6.5 months (3–24 months) from revision surgery. In patients with presumed aseptic long bone nonunion and normal CRP levels, the risk of underlying low-grade indolent infection can be as high as 26%. Patients should be made aware of this finding, which can complicate their treatment course and outcomes.

## 1. Introduction

Nonunion constitutes a major complication of the management of long bone fractures. It has been estimated that approximately 100,000 new nonunions are diagnosed every year in the USA [1]. An overall incidence of 18.94 per 100,000 population per year has recently been reported in a study involving the Scottish population [2]. Apart from the burden of the disease to the patient and the unique challenges of management posed to the treating surgeon, nonunion management is linked to significant overall -mainly indirect- associated costs [3].

The classification of nonunions to septic and aseptic is of paramount importance since it dictates management and significantly affects the outcome [4]. Infection can be clinically diagnosed when relevant signs and symptoms such as the presence of discharging sinus, increased local and systematic temperature, and redness around the nonunion site are evident. Infection can also be suspected when there is a history of the above clinical indicators. History without the presence of clinical signs of infection is significant, but in this setting, it is difficult to determine whether the nonunion should be treated as septic or not. In a recently introduced nonunion scoring system, the “previously infected or suspicion of infection” status is given one point, and the clinically “septic” status is given 4 points reflecting the importance of infective status [5]. In many circumstances, the nonunion is clustered as “aseptic” on the grounds of absent clinical signs/symptoms of infection and relevant history. However, as the possibility of a potential septic status (the so-called state of “low-grade infection”) cannot be excluded completely, investigations with inflammatory markers and imaging studies have been recommended. White blood cell count (WBC) and C-reactive protein (CRP) are commonly used to detect an acute inflammatory response in the presence of infection, but their prognostic value in the setting of nonunions has been questioned [4]. Similarly, imaging investigations such as MRI and nuclear medicine imaging studies have limited value in differentiating septic from aseptic nonunions [4]. Intraoperative positive tissue cultures from the nonunion site are currently considered a reliable indicator of an infected nonunion [6]. Interestingly, positive cultures from phenomenally “clean” or “aseptic” nonunions have not been correlated with increased risk of infection or additional surgery [7].

A reliable predictor of the infection in the presence of nonunion would be an extremely useful tool in the hand of the orthopedic traumatologists since it would provide the confidence to proceed to one or multiple stage reconstruction. The primary aim of the present study was to investigate whether raised levels of CRP and WBC, in the absence of clinical signs, are correlated with positive intraoperative tissue cultures in presumptive aseptic long-bone nonunions.

## 2. Materials and Methods

Institutional board approval was obtained for this prospective cohort study (Number: AN13-1). We identified all the long bone nonunions that underwent surgical treatment from January 2011 to December 2018. A “presumptive aseptic” long bone nonunion in an adult patient constituted the inclusion criteria. For the purpose of this study, patients with fractures with only one surgery of osteosynthesis that led to nonunion were included. A presumptive aseptic long-bone nonunion was defined as a symptomatic (presence of pain or tenderness when weight-bearing, presence of pain or tenderness on palpation or examination, and the difficulty to bear weight) long bone fracture (femur, tibia, humerus, forearm and metatarsal) that failed to unite after 9 months and/or showed lack of radiographic signs of healing progression within 3 months.

Radiographic failure to healing was assessed by two orthopedic-trauma-fellowship-trained surgeons using the radiographic union scale in tibia RUST criteria [8]. All the equivocal cases were further investigated with computerized tomography, and any disagreements were resolved by consensus of the clinicians of the study. Pediatric patients, patients with incomplete case notes and patients with a “potentially infected” nonunion were excluded from the study. Patients with a previous attempt at the surgical management of the nonunion (apart from the index surgical procedure) were also excluded. “Potentially infected” nonunions were defined as the ones with local and/or systematic signs of infection at presentation or at any point during their postoperative course. Routine demographic data (gender, age), site of injury, comorbidities, diabetes, smoking habits and alcohol consumption were recorded. Drinking of more than 14 units a week on a regular basis was used to categorize the patients to alcohol “drinkers” and “non-drinkers” [9].

Preoperatively all patients were investigated with full blood count and WBC differential (normal value: 4000–11,000 per microliter) as well as CRP (normal value: <10 mg/L). These were obtained at least one week and up to four weeks (1–4 weeks) prior to the surgical management of the nonunion. Intraoperatively, at least five tissue specimens (bone, soft tissue) were obtained from the nonunion tissue and were sent for culture and sensitivity at the microbiology department. Every effort was undertaken to obtain the deepest possible tissue samples and avoid “contamination” from the superficial soft tissues. The samples were handled according to our routine laboratory protocol and were processed for both aerobic and anaerobic cultures. Antibiotics were withheld in all cases until sufficient tissue samples were obtained. One dose of intravenous gentamycin was given in all cases after tissue harvesting intraoperatively. The type of the microorganism eventually isolated if present, and the antibiotics given were also recorded.

Infection was classified as positive if any significant growth of microorganisms was observed from bone/tissue samples sent from the theater at the time of revision surgery. This growth was either noted in the normal or enriched culture medium. The prevalence and type of microorganisms were analyzed and correlated to the preoperative blood profile of the patient (WBC and CRP). Microbiology results that commented about possible contamination were treated as negative microbiology results.

The patients with positive cultures were deemed “infected cases” and were treated with antibiotics for a minimum of at least six weeks as per microbiologist advice. Clinical and radiological follow-up at two and six weeks and thereafter at three, six and twelve months from surgery were obtained for all patients. WBC and CRP were checked whenever clinically indicated. The minimum follow-up in this study was 12 months. Patients who failed to achieve union by twelve months were followed up to solid union status.

### Statistical Analysis

Cross tabulations of potential risk factors with infection outcome were provided, including sex, age, site of the nonunion, smoker, diabetes and medical comorbidities. Continuous and count covariates (age, number of comorbidities, CRP, and WBC) were categorized using validated thresholds to enable the results to be more easily interpreted. Logistic regressions were fitted to provide unadjusted, univariable odds ratios (ORs) as well as a multivariable logistic regression, which identified a single covariate (CRP) as providing a parsimonious model. This enabled a simple confusion matrix to be provided together with a calculation of relative risk from this patient prospective cohort.

## 3. Results

A total of 105 patients (59 males and 46 females) were included in the study, with an average age of 46.76 years (range 16 to 92 years) at the time of nonunion diagnosis. There were 56 femoral nonunions and 37 tibial nonunions. Twelve patients had nonunions in other long bones (humerus, radius and metatarsal).

The median time from the index surgical procedure to the time of nonunion diagnosis was 10 months (range 9 months to 10 years). The patient who was diagnosed with a nonunion after 10 years had an asymptomatic long-standing distal femoral nonunion and was referred to our institution as a recent metalwork failure.

Table 1 illustrates the patient characteristics along with cross-tabulation of potential risk factors with infection status.

All the positive cultures revealed a mixed growth of microorganisms, with coagulase-negative *Staphylococcus* (56.4%) being the most prevalent microorganism, followed by *Staphylococcus aureus* (20.5%). *Pseudomonas*, MRSA, coliforms and micrococcus were present in the remainder of the cases (23.1%). Vancomycin, either in combination with rifampicin or clindamycin, was the most commonly used pharmacological agents in the management of the infected cases.

The odds ratios (ORs) from unadjusted analysis and after adjustment by fitting a multivariable logistic regression are shown in Table 2. Odds ratios were determined for variables, and only an elevated CRP was found to be discriminatory in predicting infection at the time of revision surgery. When the CRP (normal range <10 mg/L) alone was positive, i.e., >10 mg/L in 25 cases, positive microbiology results were found in 72% of these cases (OR 9.59 (3.35–31.77, (*p* < 0.001), multivariate analysis).

Using only CRP to predict infection, the confusion matrix is given in Table 3.

Overall, the risk of infection when CRP < 10 mg/L is 21/80 = 0.26 and risk of infection when CRP is 10 mg/L or more is 0.72. Hence the relative risk given a positive CRP test is RR = 0.72/0.26 = 2.74

Solid union was achieved in all cases after an average of 6.5 months (3–24 months) from revision surgery.

## 4. Discussion

The diagnosis of infection in the presence of long bone nonunions remains a challenge, and it is of paramount importance since the characterization of the septic status will determine the treatment strategy. In this prospective cohort study, we aimed to correlate the preoperative inflammatory markers to intraoperative cultures with the view to identify whether a certain cutoff point could serve as a reliable means to prognosticate the presence of positive cultures, i.e., infection. The results of this study indicate that higher levels of CRP (>10 mg/L) are associated with positive cultures while age, gender, site of injury, smoking, alcohol consumption, diabetes, increased number of comorbidities or increased WBC are not.

The presence of infection in the bone stimulates the bone marrow to increase the production of white blood cells in order to fight the underlying infection. Consequently, the WBC count increases. Noteworthy, the vast majority of people will produce approximately 100 billion white blood cells daily. Typically, there are 4000–11,000 WBC in every microliter of blood. Types of WBCs fighting infection include neutrophils, lymphocytes, monocytes and eosinophils. In this study, where the WBC was above 11,000 per microliter (overall 4 patients), it was found that infection was present in the cultures of 2/4 patients. However, when WBC was less than 11,000 per microliter, 37/101 patients were positive, thus as a marker, its predictive value is not reliable.

This study provides evidence that in “not obviously infected long bone nonunions”, i.e., those that are seemingly aseptic or with no real clinical evidence of infection, elevated CRP levels are associated with an increased possibility of indolent infection. CRP levels above 10 mg/L are good prognosticator of indolent infection.

In musculoskeletal trauma, elevated serum inflammatory markers (WBC, ESR, CRP) should be interpreted with caution and only after other sources of infection have been excluded [10]. Stucken et al. [11] have documented that WBC and CRP can independently predict infection on the basis of positive cultures in high-risk patients for infected nonunions. It has been shown that persistent elevation or secondary raise of CRP might be an indicator of fracture-related infection [4,12]. Nevertheless, CRP is not sensitive or specific enough to detect indolent infection in nonunions without outward clinical signs and/or symptoms [13,14]. On the other hand, there is recent evidence that in nonunions with clear signs of infection or those occurring after the management of open fractures, CRP has a superior diagnostic value compared to IL-6. Wang et al. [15] investigated the diagnostic value of the serum levels of common inflammatory markers (WBC, ESR, CRP) and IL-6 in cases of nonunions in “high-risk patients” against positive intraoperative cultures. In their study of 42 patients CRP was proven to have higher sensitivity and specificity compared with serum IL-6. The authors attributed their finding to the rapid turnover of IL-6 to normal levels compared to CRP, making it an ineffective screening tool for patients with suspected infected nonunions.

Cultures from the nonunion site have been considered to be the gold standard of the diagnosis and confirmation of infection [16]. A recent study has demonstrated that the open biopsy and culture for diagnosis of long bone post-traumatic osteomyelitis is far more accurate than the soft tissue histopathologic examination and deep wound culture [17]. Nevertheless, it has been recently shown that DNA based molecular techniques have higher sensitivity in the detection of bacterial infection compared to standard culture [18]. In our study, we tested the predictive value of WBC and CRP against positive intraoperative culture, and the subsequent management of the patients with positive cultures was based on a multidisciplinary approach with the microbiologist and infectious medicine specialists taking into account the individual profile of the patient. In our study, 37.1% of the nonunions had a positive culture, and all of them were placed in long-term (at least six weeks) antibiotics. All of the patients had a follow-up for at least one year, and none of them had a clinical relapse of the infection. This is in line with other studies that have reported positive cultures in 28.7% of cases with presumptive aseptic nonunions [19]. Amorosa et al. [19] reported that 7 patients of the total 15 with positive cultures underwent a secondary operation. *Staphylococcus* species were the most commonly isolated bacteria in the present study, a finding which is in line with other studies [7,15,18].

Our intention in this study was to focus exclusively on patients with presumptive aseptic nonunion, i.e., patients with no history of the previous infection and no symptoms and/or signs of local and/or systematic infection. We believe that this category of nonunions is the one that has the potential risk of serious complications should the infection is missed on the grounds of absent clinical signs of infection and the nonunion is treated as an aseptic one. In patients with no clinical signs of infection and normal CRP values, unforeseen positive cultures have been noted. Similarly, in a large retrospective study, Olszewski et al. [7] reported ninety-one (20%) of “unexpected” positive cultures. The authors recommended antibiotics in all cases stating that this approach is safer based on the potential implications of missing an infection with the presumption that a positive culture is a contaminant. Our treatment protocol and recommendations agree with the above study, and in our case series, all nonunions with positive cultures were treated with antibiotics, and we observed no relapse of infection and clinical union. Nonunions with negative cultures were treated without antibiotics, and no relapse of infection and uneventful healing was also observed.

To the best of our knowledge, the present study is only one in the contemporary orthopedic literature that has demonstrated that the risk of infection in patients with no clinical signs of infection and normal CRP is 21/80 patients = 0.26. In other words, patients should be informed that there is a chance of 26% to have a low-grade infection despite normal biochemical markers and no clinical signs of infection. This finding can have implications in their treatment, including (a) prolonged hospital stay until microbiological cultures and sensitivities have been completed (up to 5 days); (b) if an antibiotic-resistant strain will be isolated, then prescription of iv delivery of antibiotics for at least 2 weeks may be necessary prior to switching to oral medication. This requirement may further increase the length of hospital stay; (c) length of antibiotic treatment for up to 8 weeks; (d) risk of treatment failure and need for further surgery.

On the other hand, when CRP is elevated, the possibility of having an underlying infection is 72%.

This information should be preoperatively conveyed to the patient in the informed consent process. Although not documented in our study, positive cultures and IV antibiotic administration may potentially lead to a longer hospital stay, and this should be taken into account in planning and organizing the postoperative care of the patient.

Surgeons should feel confident that elevated CRP can reliably predict positive cultures in presumptive aseptic nonunions in almost three-quarters of patients managed. Antibiotics should be administered during the immediate postoperative period, and no ‘antibiotic gap should exist waiting for finalization of tissue cultures.

This study has limitations. First, we only used positive cultures to define infection. Newer techniques such as sonication, fluorescent in situ hybridization and polymerase chain reaction have not been used and could possibly be incorporated in another future study. Interestingly, in a recent systematic review of the literature investigating the accuracy of tissue and sonication fluid sampling for the diagnosis of fracture-related infection, the authors reported that there is limited evidence that sonication fluid culture could be a useful adjunct to conventional tissue culture, but no strong evidence that it is superior or can replace tissue culture [19]. Moreover, the authors found that for molecular techniques and histopathology, the evidence was even less clear. The authors concluded that standardization of laboratory protocols and consistent and uniform diagnostic criteria should be implemented in the clinical setting.

Second, we could not comment on the severity and type of initial injury, especially in open fractures, since most of the cases were initially treated in different institutions and were referred to us only when a nonunion had been diagnosed. Third, nonunions from different anatomical sites were included in the study (femur, tibia, humerus, radius and metatarsal). However, this is a common drawback of studies related to fracture nonunions.

Strengths of the study include the findings from a consecutive series of patients from one institution (large tertiary referral center) and the reasonably large number of patients studied.

In conclusion, the herein study of 105 consecutive patients provides evidence that elevated CRP levels and not WBC can be used as a discriminator for the detection of low-grade infection in otherwise aseptic long bone nonunions. However, the risk of infection can be as high as 26% in patients with normal CRP levels, and this finding should be explained to them during the consent process of revision surgery.

## Figures and Tables

**Table 1 jcm-10-00425-t001:** Patient demographics, investigations and comorbidities.

		Stratified by Microbiology Result		
		Negative	Positive	*p*-Value
n		66	39	
Gender (%)	Female	30 (45.5)	16 (41.0)	0.812
	Male	36 (54.5)	23 (59.0)	
Age (%)	Under 65	55 (83.3)	29 (74.4)	0.391
	65 and over	11 (16.7)	10 (25.6)	
Site (%)	Femur	34 (51.5)	22 (56.4)	0.753
	Other	7 (10.6)	5 (12.8)	
	Tibia/Fibula	25 (37.9)	12 (30.8)	
Comorbid (%)	No	13 (19.7)	12 (30.8)	0.294
	Yes	53 (80.3)	27 (69.2)	
Smoker (%)	No	54 (81.8)	33 (84.6)	0.921
	Yes	12 (18.2)	6 (15.4)	
Alcohol (%)	No	55 (83.3)	36 (92.3)	0.312
	Yes	11 (16.7)	3 (7.7)	
Diabetes (%)	No	60 (90.9)	36 (92.3)	0.999
	Yes	6 (9.1)	3 (7.7)	
CRP (%)	Below 10 mg/L	59 (89.4)	21 (53.8)	<0.001
	Over 10 mg/L	7 (10.6)	18 (46.2)	
WBC (%)	Below 11,000 per microliter	64 (97.0)	37 (94.9)	0.988
	Over 11,000 per microliter	2 (3.0)	2 (5.1)	

CRP: C-reactive protein; WBC: white blood cell count.

**Table 2 jcm-10-00425-t002:** Univariate and multivariate analysis of risk factors predicting infection.

		OR (Univariable)	OR (Multivariable)
Gender	Female	-	-
	Male	1.20 (0.54–2.70, *p* = 0.659)	1.68 (0.63–4.75, *p* = 0.309)
Age 65	Under 65	-	-
	65 and over	1.72 (0.65–4.57, *p* = 0.270)	2.59 (0.72–9.70, *p* = 0.148)
Site	Femur	-	-
	Other	1.10 (0.29–3.90, *p* = 0.878)	1.13 (0.25–4.80, *p* = 0.866)
	Tibia/Fibula	0.74 (0.30–1.76, *p* = 0.502)	0.66 (0.21–1.96, *p* = 0.460)
Smoker	No	-	-
	Yes	0.82 (0.26–2.32, *p* = 0.714)	0.37 (0.11–1.16, *p* = 0.091)
Alcohol	No	-	-
	Yes	0.42 (0.09–1.44, *p* = 0.202)	0.29 (0.04–1.43, *p* = 0.154)
CRP	Under 10 mg/L	-	-
	10 mg/L or over	**7.22** (2.74–20.93, *p* < 0.001)	10.45 (3.54–35.87, *p* < 0.001)
WBC	Below 11,000 per microliter	-	-
	Above 11,000 per microliter	1.73 (0.20–14.10, *p* = 0.592)	0.68 (0.04–10.06, *p* = 0.781)

**Table 3 jcm-10-00425-t003:** Confusion matrix for C-reactive protein (CRP) prediction of infection.

CRP Level	Positive Microbiology	Negative Microbiology	Total
CRP > 10 mg/L	18	7	25
CRP < 10 mg/L	21	59	80

Positive predictive value PPV = 18/25 = 0.72, so 72% and 95% CI is (54%, 85%). Negative predictive value NPV = 59/80 = 0.74, so 74% and 95% CI is (68%, 79%).

## Data Availability

The data presented in this study are available on request from the corresponding author. The data are not publicly available due to ethical restrictions.

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
