# Peer review of "Can CRP Levels Predict Infection in Presumptive Aseptic Long Bone Non-Unions? A Prospective Cohort Study"

_jcm, 2021, doi:10.3390/jcm10030425_

Round 1

Reviewer 1 Report

Overall

Well written paper on an interesting topic. Overall study design and methodology seems reasonable but currently the details of the multi-variable analysis are unclear and potentially may need to be modified after clarification. Are there significant difference in demographics/other confounders such as smoking between the CRP >10 and <10 groups / are these confounders appropriately addressed in the multi variable analysis.

Furthermore although WBC count was found to be a non-significant predictor, slightly more discussion of this is warranted in the paper

Introduction

Line 56 – “Investigations of a possible ‘septic status’ the so-called state of ‘low grade infection’ with inflammatory markers and preoperative cultures from non-union tissue and imaging studies have been recommended.” Awkward wording, reword.

Line 59 – “detect acute inflammatory response in the presence infection” – missing “presence of infection”

Methods

Line  127- Define threshold values for crp, WBC, ect in your methods section

Line 129 – Here you say this is a retrospective, title states it is a prospective study, please clarify study design.

Results

I am unclear what variables were used in the multi-variable analysis based off the methods section

Why was 7 used as the threshold for WBC? This seems very low?

Please make sure to provide units throughout the paper, they are missing in a number of areas

Discussion

Line 222 – “positive ‘surprise’ cultures have been noted.” Reword.

Line 245 – Would be interesting to note how often negative cultures provide false negatives compared to these newer techniques

Author Response

Overall

Well written paper on an interesting topic. Overall study design and methodology seems reasonable but currently the details of the multi-variable analysis are unclear and potentially may need to be modified after clarification. Are there significant difference in demographics/other confounders such as smoking between the CRP >10 and <10 groups / are these confounders appropriately addressed in the multi variable analysis.

Furthermore, although WBC count was found to be a non-significant predictor, slightly more discussion of this is warranted in the paper.

Reply: We thank the reviewer for the points raised. We have taken this into consideration and have responded appropriately in the manuscript. Particularly, for the WBC count we added the following paragraph in the discussion:

‘The presence of infection in the bone stimulates the bone marrow to increase the production of white blood cells in order to fight the underlying infection. Consequently, the WBC count increases.  Noteworthy, the vast majority of people will produce approximately 100 billion white blood cells daily. Typically, there are 4,000-11,000 WBC in every microliter of blood. Types of WBCs fighting infection include neutrophils, lymphocytes, monocytes and eosinophils.  In this study, patients where the WBC was above 11,000 (overall 4 patients) it was found that infection was present in the cultures of 2/4 patients. However, when WBC was less than 11,000 37/101 patients were positive, thus as a marker its predictive value is not reliable.’

Introduction 

Line 56 – “Investigations of a possible ‘septic status’ the so-called state of ‘low grade infection’ with inflammatory markers and preoperative cultures from non-union tissue and imaging studies have been recommended.” Awkward wording, reword.

Reply: Sentence has been revised as follows:

‘However, as the possibility of a potential septic status (the so-called state of ‘low grade infection’) cannot be excluded completely, investigations with inflammatory markers and imaging studies have been recommended’.

Line 59 – “detect acute inflammatory response in the presence infection” – missing “presence of infection”

Reply: Error corrected accordingly.

Methods

Line 127- Define threshold values for crp, WBC, ect in your methods section

Reply: Values added accordingly.

Line 129 – Here you say this is a retrospective, title states it is a prospective study, please clarify study design.

Reply: Thank you. This has been addressed accordingly.

Results

I am unclear what variables were used in the multi-variable analysis based off the methods section:

Reply: Variables used in the multi-variable analysis included sex, age, smoking, alcohol, CRP, WBC, anatomical site, medical comorbidities.  

Why was 7 used as the threshold for WBC? This seems very low?

Reply: We than the reviewer for the helpful comment. We have rechecked the data and redone the calculations setting the threshold of WBC to 11,000. This has been included in both revised Table 1 and Table 2.

Please make sure to provide units throughout the paper, they are missing in a number of areas

Reply: This has been corrected throughout the paper accordingly.

Discussion

Line 222 – “positive ‘surprise’ cultures have been noted.” Reword.

Reply: The sentence has been revised as follows: ‘In patients with no clinical signs of infection and normal CRP values positive unforeseen cultures have been noted’.  

Line 245 – Would be interesting to note how often negative cultures provide false negatives compared to these newer techniques

Reply. We than the reviewer for the interesting point raised. However, currently there is limited evidence available to provide some useful data. Consequently, the following paragraph was included:

‘Interestingly, in a recent systematic review of the literature investigating the accuracy of tissue and sonication fluid sampling for the diagnosis of fracture related infection, the authors reported that there is limited evidence that sonication fluid culture could be a useful adjunct to conventional tissue culture, but no strong evidence that it is superior or can replace tissue culture. Moreover, the authors found that for molecular techniques and histopathology the evidence was even less clear. The authors concluded that standardisation of laboratory protocols and consistent and uniform diagnostic criteria should be implemented in the clinical setting’.

Reviewer 2 Report

line 39- I assume 100,000? instead of 100.000

line 26- ORs- odd ratios needs to be spelled at the first time, it is spelled out in line 150 instead 

line 233-234- The 0.26 risk of infection for the normal CRP should be elaborated more. While the study shows that CRP has a high risk of having infection. 0.26 is not low. Should surgeons being using antibiotics be used for all nonunions like the Olszewski article recommends? I'm curious more on the authors thoughts about this more than just informing the patient they have a 26% chance of infection. 

Did authors retrospectively try to look back at this subset of negative CRP but positive cultures and see if there was any other significant variable that may have influenced the results. 

Did authors get metabolic labs on the all their nonunions and rule out other causes of nonunion during the aspetic work up?

Author Response

line 39- I assume 100,000? instead of 100.000

Reply: Thank you. This has been corrected.

line 26- ORs- odd ratios needs to be spelled at the first time, it is spelled out in line 150 instead 

Reply: This has now been corrected.

line 233-234- The 0.26 risk of infection for the normal CRP should be elaborated more. While the study shows that CRP has a high risk of having infection. 0.26 is not low. Should surgeons being using antibiotics be used for all nonunions like the Olszewski article recommends? I'm curious more on the authors thoughts about this more than just informing the patient they have a 26% chance of infection. 

 Reply: the following sentence has been added in the manuscript:

‘This finding can have implications in their treatment including:

  1. a) Prolonged hospital stay until microbiological cultures and sensitivities have been completed (up to 5 days).
  2. b) If antibiotic resistant strain will be isolated, then prescription of iv delivery of antibiotics for at least 2 weeks may be necessary prior to switching to oral medication. This requirement may further increase the length of hospital stay.
  3. c) Length of antibiotic treatment for up to 8 weeks.
  4. d) Risk of treatment failure and need for further surgery.

Did authors retrospectively try to look back at this subset of negative CRP but positive cultures and see if there was any other significant variable that may have influenced the results. 

Reply: We thank the author for their comment. We indeed look into the natural history of the non-union, injury pattern, age, sex and medical comorbidities of patients but we did not identify any association of interest to report.

Did authors get metabolic labs on all their non-unions and rule out other causes of non-union during the aseptic work up?

Reply: We have a standardised protocol in our institution where all the important parameters that have been implicated in the pathogenesis of non-union are routinely investigated including patient metabolic profile (bone chemistry, thyroid function test, Vitamin D, blood sugar, etc.). All patients underwent investigations. Nonetheless, the focus of the herein study was to investigate whether raised preoperative levels of CRP and WBC, in the absence of clinical signs, are correlated with positive intraoperative tissue cultures in presumptive aseptic long-bone non-unions.